# Quinacrine-Induced Autophagy in Ovarian Cancer Triggers Cathepsin-L Mediated Lysosomal/Mitochondrial Membrane Permeabilization and Cell Death

**DOI:** 10.3390/cancers13092004

**Published:** 2021-04-21

**Authors:** Prabhu Thirusangu, Christopher L. Pathoulas, Upasana Ray, Yinan Xiao, Julie Staub, Ling Jin, Ashwani Khurana, Viji Shridhar

**Affiliations:** 1Department of Experimental Pathology and Medicine, Mayo Clinic, Rochester, MN 55905, USA; thirusangu.prabhu@mayo.edu (P.T.); pathoulas@uchc.edu (C.L.P.); ray.upasana@mayo.edu (U.R.); Xiao.Yinan@mayo.edu (Y.X.); staub.julie@mayo.edu (J.S.); jin.ling@mayo.edu (L.J.); khurana.ashwani@mayo.edu (A.K.); 2Department of Obstetrics and Gynecology, The Second Xiangya Hospital, Central South University, Changsha 410008, China

**Keywords:** quinacrine, autophagy, CTSL, LMP, MOMP, ovarian cancer

## Abstract

**Simple Summary:**

Ovarian cancer (OC) is the most common cause of cancer-related deaths among women worldwide, and its incidence has been increasing and has continued to prove resistant to a variety of therapeutics. This observation is principally disturbing given the amount of money invested in identifying novel therapies for this disease. A comparatively rapid and economical pipeline for identification of novel drugs is drug repurposing. We reported earlier that the antimalarial drug Quinacrine (QC) also has anticancer activity and here we discovered that QC significantly upregulates cathepsin L (CTSL) and promoting autophagic flux in ovarian cancer. QC-induced CTSL activation promotes lysosomal membrane permeability resulting in active CTSL release into the cytosol, which promotes Bid cleavage, mitochondrial membrane permeability, cytochrome-c release and cell death in both in-vitro and in-vivo models. Therefore, QC is a promising candidate for OC treatment.

**Abstract:**

We previously reported that the antimalarial compound quinacrine (QC) induces autophagy in ovarian cancer cells. In the current study, we uncovered that QC significantly upregulates cathepsin L (CTSL) but not cathepsin B and D levels, implicating the specific role of CTSL in promoting QC-induced autophagic flux and apoptotic cell death in OC cells. Using a Magic Red^®^ cathepsin L activity assay and LysoTracker red, we discerned that QC-induced CTSL activation promotes lysosomal membrane permeability (LMP) resulting in the release of active CTSL into the cytosol to promote apoptotic cell death. We found that QC-induced LMP and CTSL activation promotes Bid cleavage, mitochondrial outer membrane permeabilization (MOMP), and mitochondrial cytochrome-c release. Genetic (shRNA) and pharmacological (Z-FY(tBU)-DMK) inhibition of CTSL markedly reduces QC-induced autophagy, LMP, MOMP, apoptosis, and cell death; whereas induced overexpression of CTSL in ovarian cancer cell lines has an opposite effect. Using recombinant CTSL, we identified p62/SQSTM1 as a novel substrate of CTSL, suggesting that CTSL promotes QC-induced autophagic flux. CTSL activation is specific to QC-induced autophagy since no CTSL activation is seen in ATG5 knockout cells or with the anti-malarial autophagy-inhibiting drug chloroquine. Importantly, we showed that upregulation of CTSL in QC-treated HeyA8MDR xenografts corresponds with attenuation of p62, upregulation of LC3BII, cytochrome-c, tBid, cleaved PARP, and caspase3. Taken together, the data suggest that QC-induced autophagy and CTSL upregulation promote a positive feedback loop leading to excessive autophagic flux, LMP, and MOMP to promote QC-induced cell death in ovarian cancer cells.

## 1. Introduction

Ovarian cancer (OC) is the leading cause of death in women with gynecological malignancy, resulting in an estimated 14,070 deaths in 2018 [1]. Although patients initially respond to platinum-based therapies [2,3], a significant majority of patients are prone to relapse and poor prognosis [1,4,5]. The success rate for de novo ovarian cancer drugs from Phase I trial to U.S. Food and Drug Administration and Likelihood of Approval was very low (4.6%) from 2003 to 2011 [6]. Therefore, drug repurposing has developed into an attractive approach for discovering new ovarian cancer therapeutics, since repurposed drugs have approved pharmaceuticals with known pharmacokinetic and safety profiles [7,8,9]. The acridine derivative quinacrine (QC) has historically been used for malaria prophylaxis and treatment but has recently been found to have new indications as an anticancer agent [10]. QC has been reported to have cytotoxic potential in many different cancer types while exhibiting limited toxicity toward normal cells [11,12,13,14,15,16,17,18]. Furthermore, QC has been shown to have multiple anticancer mechanisms and improve the cytotoxicity of other therapeutic agents (reviewed in [19]).

Previous work from our lab has shown that QC induces autophagy in OC cells and sensitizes chemoresistant cell lines to carboplatin and autophagic degradation of Skp2, which promotes p21 expression [18,20], but the molecular mechanism of QC-exhibited autophagy is still unclear. Interestingly, recent work by the Karch group demonstrated a link between autophagic cell death and lysosomal membrane permeability (LMP) [21]. Lysosomes are acidic organelles containing high amounts of hydrolytic enzymes that are responsible for degrading macromolecules received from a variety of membrane-trafficking pathways including autophagy [22,23]. LMP releases lysosomal hydrolases into the cytoplasm, leading to aberrant degradation of cellular components. Importantly, the downstream pathways activated by LMP can be influenced by a variety of factors and, while complete lysosomal rupture can lead to necrosis, partial rupture results in the selective release of cathepsin proteases, which can induce both caspase-dependent and caspase-independent cell death [24,25,26,27,28]. Several cathepsin proteases that have been implicated in cell death include cathepsin B (CTSB), cathepsin D (CTSD), and cathepsin L (CTSL) [24]. Although cathepsins have long been thought to be unstable at neutral pH, they have been found to remain active at neutral pH for varying amounts of time ranging from minutes (cathepsin L [29]) to hours (cathepsin S [30]). Cleavage of Bid by cathepsins leads to mitochondrial outer membrane permeabilization (MOMP), and subsequent release of cytochrome-c (cyt-c) into the cytoplasm [27,28,31] demonstrates the importance of lysosomal–mitochondrial crosstalk in LMP-induced apoptosis [24].

Given the potential role of cathepsins, LMP, and MOMP in autophagic and apoptotic cell death, we decided to investigate the role of cathepsins in the crosstalk between LMP and MOMP in QC-induced autophagy to promote cytotoxicity. Our study revealed that QC specifically upregulated CTSL activity (and not CTSB or D) more so in chemoresistant OC cells, warranting further examination of this phenomenon. We uncovered that QC-induced CTSL upregulation promotes autophagic flux, LMP, and MOMP, resulting in autophagic and caspase-dependent cell death in ovarian cancer. The study findings offer new insight into the cytotoxic mechanisms of QC and demonstrate the potential for targeting lysosomes in ovarian cancer.

## 2. Results

### 2.1. QC Induces Cytotoxicity and Lysosomal Membrane Permeability (LMP) in Ovarian Cancer Cells

Given that QC is highly concentrated in lysosomes and LMP plays a role in autophagic and apoptotic cell death [21,24,32], we investigated whether QC induces LMP using LysoTracker (Thermofisher, Waltham, MA, USA) red live cell stain. Initial exposure to both 5 and 10 µM QC for 3, 6, 12 and 24 h in C13 and HeyA8-MDR cells showed loss of red fluorescent signal, indicating induction of LMP. QC (5 µM) was effective in inducing LMP in C13 and HeyA8-MDR cells as early as 3 and 6 h, respectively (Figure 1a,c). Figure 1b,d shows the quantitation of the red signal loss for C13 and HeyA8-MDR cells, respectively. Cytotoxic assessment of QC was validated by MTT assay (Methods Section 4.2), which showed cell viability was inhibited as early as 24 h of QC treatment at IC_50_ ~ 2.5–5 µM. QC showed an enhanced sensitivity at 48 h treatment (Appendix A).

### 2.2. QC Actuates CTSL Expression and Its Activity

To examine whether QC-induced LMP had an effect on lysosomal cathepsins, we analyzed the effect of QC on CTSB, CTSD, and CTSL as these cathepsins are associated with LMP and cell death. Western blot analysis showed that in C13 and HeyA8MDR cells, QC specifically upregulated CTSL but not CTSB or CTSD (Figure 1e and Appendix A). q-PCR and immunoblot analysis showed that both transcriptional (Figure 1f) and translational CTSL (>2 fold higher) were higher in the chemoresistant C13 (resistant to carboplatin/cisplatin) than in its chemo-sensitive counterpart OV2008 cells upon QC treatment (Figure 1g, panel 2 and Appendix A). C13 cells also displayed higher induction in the cleaved PARP, LC3BII and degradation of p62/SQSTM1 compared to OV2008 cells (Figure 1g, panels 1, 5 and 4, respectively).

To further validate the effect of QC on the activity of CTSL we used the Magic Red^®^ (ImmunoChemistry Technologies, Bloomington, IN, USA) CTSL live cell activity assay. When CTSL is active, the Magic Red substrate MR-FR2 is cleaved and the cresyl violet fluorophore will become fluorescent and, therefore, the increase in signal intensity reflects increased CTSL activity within the cell. A dose-dependent increase in CTSL activity was found upon QC treatment of C13 and HeyA8MDR cells (Figure 1h,j, panels 1–4), which was completely attenuated in the presence of Z-FY(tBU)-DMK (Figure 1h,j, panel 5), respectively. Quantification of the fold increase indicated that treatment of C13 and HeyA8MDR cells with 5 and 10 µM QC increased CTSL activity approximately 10-fold and 12-fold, respectively (Figure 1i,k), leading us to further examine the role of CTSL in QC-induced cell death.

### 2.3. QC-Induced Cytotoxicity Is Mediated by CTSL Activity

To understand the role of lysosomal cathepsins in QC-mediated cytotoxicity, HeyA8-MDR cells were treated with Pepstatin A (CTSD inhibitor), CA-074Me (CTSB inhibitor) and Z-FY (tBU)-DMK (CTSL inhibitor) at indicated concentrations in the presence and absence of 5 µM QC for 24 h and assessed by clonogenic assays, Annexin V/PI staining and Western blot. The colony-forming ability of cells treated with 5 µM QC was significantly inhibited compared to untreated cells (Figure 2a,d,g + QC). However, the addition of Z-FY (tBU)-DMK rescued colony formation (Figure 2g,h + QC + CTSL inhibitor), whereas the addition of CTSD or CTSB inhibitors did not (Figure 2a,b,d,e). Consistent with these data, immunoblot also showed a reduction in cleaved PARP levels when the cells were treated with CTSL inhibitor in combination with QC compared to QC treatment alone (Figure 2i, lane 4 compared to lane 2). In contrast, the combination treatment of QC with CTSD and CTSB inhibitors had no effect on cleaved PARP levels (Figure 2c,f). Further validation showed a significant reduction in the levels of cleaved caspases 9/3 when treated with CTSL inhibitor in combination with QC compared to QC alone in both C13 and HeyA8-MDR cells (Figure 2j) (for densitometry analysis refer to Appendix A). Furthermore, QC treatment alone showed an increase in Annexin V (early apoptosis) (~23.5% in C13 and ~32.8% in HeyA8-MDR). However, treatment with QC in the presence of CTSL inhibitor showed a reduction in early apoptotic events with up to ~10.4% in C13 and ~11.4% in HeyA8-MDR cells (Figure 2k). Taken together, these results demonstrated that CTSL, but not CTSB or CTSD, mediates QC-induced cytotoxicity.

### 2.4. QC-Induced CTSL Activation Is Dependent on Autophagy and Vice Versa

To further support the data that QC-induced ovarian cancer cell death is dependent on autophagy, we performed FACS analysis on C13 non-targeted control transduced (NTC) and ATG5 knockdown (KD) cells after treatment with QC ± caspase inhibitor (Z-VAD-FMK). We found that ATG5 KD significantly attenuated QC-induced apoptosis similar to treating NTC cells with caspase inhibitor, indicating that autophagy is essential for QC-induced apoptosis (Figure 3a, 33.8% of cell death in QC treated in NTC whereas no apoptotic cell death in the presence of caspase inhibitor in NTC or ATG5 KD cells). Consistent with this data, ectopic expression of CTSL in OV2008 (low basal CTSL) showed increased early (38.2%) and late (2.6%) apoptotic events in the presence of QC compared to that of 24.9% and 2.4% of vector control cells, respectively. However, treatment with Z-VAD-FMK resulted in significant reduction in apoptosis (Figure 3b).

Given the importance of autophagy in QC-induced apoptosis and cell death, we chose to examine the role of QC-induced autophagy in promoting CTSL upregulation. Here, we proved that inhibition of autophagy pharmacologically with 3-methyladenine or chloroquine attenuates QC-induced CTSL upregulation in both HeyA8-MDR and OVCAR5 cells (Figure 3c,d, Lanes 4–12, top panel), respectively. Inhibition of autophagic flux under this treatment condition was further confirmed by restoration of p62 expression in the cells, resulting in reduced cleaved PARP (Figure 3c,d, panels 3 and 2, respectively) (for densitometry analysis refer to Appendix A).

To further confirm the induction of autophagic flux, we monitored the fluorescence of GFP-RFP-LC3B/or p62 construct in both the C13 and HeyA8 MDR cells upon treatment with QC for 24 h. The conversion of early autophagosome or p62 inclusion body into autolysosome was monitored by formation of green (GFP+ and RFP−) to yellow puncta (GFP+ and RFP+) and increased red puncta (GFP− and RFP+) in the cells (Figure 4a). Confocal images of the transiently transfected GFP-RFP-LC3B/or p62 construct in C13 (Figure 4b,c) and HeyA8-MDR (Figure 4d,e) cells showed induction of autophagic flux through the formation of both yellow/orange (GFP/RFP) and increased red-RFP puncta in 5 µM QC-treated cells compared to the untreated cells (no autophagosome or p62 autophagosomal inclusion bodies), respectively. Furthermore, OV2008 cells with ectopic expression of CTSL transfected with GFP-RFP-LC3B/or p62 showed increased autophagic flux compared to OV2008-vector cells (Figure 4f,g, and respective graphs depict the increase of autolysosome-RFP puncta per cell). However, knockdown of CTSL in C13 cells showed reduced formation of autophagosomal bodies even in the presence of QC (Figure 4h,i), thus confirming the role of CTSL in inducing autophagic flux.

### 2.5. CTSL Promotes Autophagic Flux by Degrading p62

Lysosomal cathepsins play an important role in promoting autophagic flux by degrading proteins transported to the lysosome [33], therefore we examined the effect of CTSL knockdown and overexpression on QC-induced autophagy. For this we generated stable CTSL KD clones of C13 and OVCAR5 cells (which have high basal CTSL expression) using two different shRNA and CTSL-overexpressing clones in OV2008 and OVCAR7 cells (which have low basal CTSL expression) and treated with 0, 2.5, 5 and 10 µM QC. We found that CTSL knockdown significantly decreased QC-induced LC3BII and prevented degradation of p62 protein compared to C13-NTC cells, suggesting that CTSL knockdown (Figure 5a, top panel) impairs QC-induced autophagic flux, as evidenced by p62 and LC3B levels (Figure 5a, panels 2 and 3) (Appendix A). In contrast, overexpression of CTSL (Figure 5b, top panel) led to degradation of p62, increased LC3BII level when compared to OV2008-vector cells. Similar results were obtained with CTSL KD in high-grade serous OVCAR5 cells and in CTSL-overexpressed high-grade serous OVCAR7 cells (Figure 5c,d), respectively (for densitometric analysis refer to Appendix A).

To understand the regulation of p62 by CTSL, immunofluorescence analysis in HeyA8-MDR cells treated with QC showed a time-dependent increase in the CTSL expression (Figure 5e, green panels) with a reduction in cytoplasmic p62 levels (Figure 5e, red panels). To further validate, Magic Red was used to validate increased CTSL activity in CTSL-overexpressing OV2008 cells (Red signal) associated with significantly low levels of cytoplasmic p62 protein compared to the vector control cells (Figure 5f, green signal). To delineate that the degradation of p62 was directly mediated by upregulation of CTSL, whole cell lysates from HeyA8-MDR cells were incubated with recombinant human CTSL and CTSB protein in both concentration-dependent (0.0, 0.2, 0.5 and 1.0 µg/sample for 2 h) and/or time-dependent manner (0.5 µg for 1, 2 and 3 h). Immunoblot analysis showed degradation of p62 protein was only mediated by rCTSL in both concentration and time-dependent manner (Figure 5g,h) but not by rCTSB (Figure 5i) (for densitometry analysis refer to Appendix A).

Furthermore, to evaluate CTSL-mediated p62 cleavage pattern, myc-tagged p62 was overexpressed in Hek293T cells, and lysates were incubated with rCTSL or rCTSB at 0.0, 0.1, 0.25 µg/sample for 1 h. The results showed that in the presence of rCTSL, ectopically expressed p62 was cleaved at two sites, resulting in two bands of approximately ~45 and 42 KD fragments, compared to rCTSL untreated and/or with rCTSB (Appendix A). CTSL has mainly endopeptidase activity and preferentially cleaves peptide bonds Phe–Arg or Arg–Arg with aromatic and hydrophobic residues [34], which supports the experimental results, as shown in Appendix A. Taken together our data suggest for the first time that QC induces a positive feedback loop where QC-induced autophagy upregulates CTSL, which goes on to further promote autophagic flux by degrading p62.

### 2.6. CTSL Mediates QC-Induced LMP, Bid Cleavage and MOMP

Autophagy and lysosomal cathepsins have previously been shown to play a role in LMP [21,35,36,37]. LMP induced by TNF-alpha has been shown to be dependent on CTSB [38] and the cationic amphiphilic drug resveratrol has been shown to induce LMP in cervical cancer cells that are dependent on autophagy and CTSL [35]. Our data in Figure 1a,b clearly indicate that QC induces LMP, therefore we hypothesized that induced upregulation of CTSL may play a role in QC-mediated LMP. To validate the role of QC-upregulated CTSL in inducing LMP, we treated CTSL-knockdown C13 cells and CTSL-overexpressing OV2008 cells with 5 μM QC for 12 and 24 h and determined the extent of LMP by analyzing the loss of LysoTracker red using confocal microscopy. Results showed that knockdown of CTSL significantly reduced the loss of LysoTracker red signal induced by QC (Appendix A) in a time-dependent manner. In contrast, ectopic overexpression of CTSL in OV2008 cells led to a significant increase in the loss of LysoTracker red signal after QC treatment (Appendix A). Collectively, these results suggested that QC-upregulated CTSL may have a critical role in QC-mediated LMP to induce the cell death.

Lysosomal cathepsins released after LMP induce MOMP and cyt-c release by cleaving the Bcl-2 family member Bid [27]; we sought to investigate whether QC induces MOMP and the role of CTSL in promoting the MOMP. Mitochondrial release of cyt-c into the cytoplasm indicates MOMP induction. However, to further affirm whether QC promotes MOMP, MitoTracker Red dye (Thermofisher, Waltham, USA) was used to assess mitochondrial outer membrane integrity, and QC treatment led to significant loss of MitoTracker Red stain compared to untreated C13 cells, affirming that QC induces MOMP. Interestingly, rescue of red signal was evident where cells were treated with QC in the presence of Z-FY(tBU)-DMK (Figure 6a,b). Furthermore, we found that QC-induced MOMP was mediated by CTSL as QC treatment in CTSL KD C13 cells prevented QC-induced loss of MitoTracker Red (Figure 6c,d). To further confirm CTSL-upregulated OV2008 cells showed a significant loss of MitoTracker Red signal compared to control cells, however, treatment with CTSL inhibitor prevented the MOMP in the overexpressed cells (Figure 6e,f). Retention of red signal was also obtained when the CTSL-overexpressed cells were treated with QC in the presence of Z-FY(tBU)-DMK (Figure 6e,f), which confirmed the role of CTSL in mediating QC-induced MOMP in ovarian cancer cells.

Furthermore, QC treatment of CTSL KD C13 and OVCAR 5 cells showed a significant reduction in tBid levels (a specific substrate for CTSL and apoptotic factor) and cyt-c release and attenuated expression levels of active caspases-3 and cleaved PARP levels (Figure 6g,h, panels 3 and 4, respectively) compared to NTC transduced cells. In contrast, overexpression of CTSL in OV2008 and OVCAR7 showed increased cyt-c release accompanied by generation of tBid, an increase in active caspase 9/3 expression and cleaved PARP after QC treatment, compared to vector cells (Figure 6i,j, panels 1, 2 and panels 3–5, respectively, for densitometric analysis refer to Appendix A). Collectively these results supported the notion that QC-induced CTSL promotes MOMP, leading to caspase-mediated cell death.

To further confirm the role of CTSL in QC-induced cell death, we treated the CTSL KD C13 and overexpressed OV2008 cells with QC and analyzed the effect on the anti-apoptotic and the pro-apoptotic protein markers. Immunoblot analysis showed that QC induced degradation of the anti-apoptotic proteins Mcl1 and Bcl-w and promoted the pro-apoptotic proteins like Bim and Bad, which are impaired in CTSL KD C13 cells compared to the NTC cells (Appendix A). Moreover, the CTSL-overexpressed OV2008 cells showed the opposite effect on treatment with QC for 24 h (Appendix A), which confirmed that QC-mediated CTSL induction promotes apoptotic cell death in ovarian cancer cells.

### 2.7. QC Upregulates CTSL and Promotes Ovarian Cancer Cell Death In Vivo

We previously reported that QC treatment of mouse-derived xenografts of HeyA8-MDR cells was shown to be effective in reducing tumor weight and ascitic fluid formation (Figure 7a–d in Ashwani et al. [18]) with increased formation of autophagosomes and autolysosomes, and these effects were dramatic when QC was combined with carboplatin, but QC exhibited a molecular mechanism that was still unclear. Hence, we tested four each of control and QC-treated xenografts to check if QC treatment induced active CTSL in vivo. As shown in Figure 7a, there was a dramatic induction of CTSL upon QC treatment in vivo. This correlated with degradation of p62, upregulation of LC3BII and increase in cyt-c, cleaved Bid, caspase-3 and cleaved PARP compared to untreated tumors (for densitometry analysis refer to Appendix A), effectively mimicking the results we obtained in vitro. Taken together these data provide strong evidence that CTSL mediates QC-induced autophagic and apoptotic cell death in vivo. Figure 7b shows the model figure of QC-activated CTSL promoting autophagy-mediated LMP and MOMP-dependent cell death.

## 3. Discussion

LMP can be triggered by a wide variety of stimuli including several antimalarial drugs [35,36]. The antimalarial drugs hydroxychloroquine (HCQ) and chloroquine (CQ) increase lysosomal pH leading to lysosomal dysfunction and autophagy inhibition. Now although HCQ has long been established as an autophagy inhibitor, work by the Boya team showed that HCQ induces LMP and MOMP, resulting in hallmark autophagic and apoptotic cell death [39,40].

In the present study, we showed that QC induces autophagy, LMP, and MOMP, leading to ovarian cancer cell death. We found that QC-induced cytotoxicity was mediated by the upregulation and activation of the cysteine protease CTSL (Figure 2a–i). Furthermore, we discovered that CTSL mediates QC-induced autophagic and apoptotic cell death in OC cells (Figure 2j,k and Figure 3a–e). Interestingly, a recent report highlighted that treatment with CQ and HCQ-induced LMP mediated cell death with and without cathepsin activation, respectively [39,40]. Taken together, recent work using CQ, HCQ, and our work using QC have suggested that the anticancer properties of these antimalarial drugs may be closely linked to their ability to induce LMP (Figure 1 and Figure 3e).

Lysosomal–mitochondrial crosstalk can play an important role in mediating cell death following LMP, as the selective release of cathepsins after LMP can promote MOMP and cyt-c release by cleaving the Bcl-2 family member Bid [27,28,31]. Therefore, after showing QC-induced LMP (Figure 1a–d) and CTSL activation (Figure 1e–k), we examined whether CTSL mediates QC-induced apoptosis by promoting Bid cleavage, cyt-c release, and MOMP. In this study, we showed that CTSL inhibition markedly reduced QC-induced Bid cleavage to tBid, cyt-c release and MOMP, whereas ectopic expression of CTSL had the opposite effect (Figure 6a–j). CTSL has also been shown to degrade anti-apoptotic proteins such as XIAP, Mcl-1, Bcl-w, and Bcl-xL, leading us to analyze whether QC-induced LMP and CTSL activation promoted the degradation of anti-apoptotic Bcl-2 family members [31]. Here, we showed that QC promoted the degradation of Mcl-1 and Bcl-w, and inhibition of CTSL rescued anti-apoptotic protein degradation (Appendix A). We further showed that CTSL knockdown or pharmacological inhibition attenuated QC-induced cleaved PARP, cleaved caspase-3, and cleaved caspase-9 levels, which are indicators of apoptosis (Figure 3a,b and Figure 6). Overall, these data confirmed that CTSL release after LMP promoted QC-induced MOMP and apoptosis and highlighted the importance of lysosomal–mitochondrial crosstalk in QC-induced cell death.

CTSL has also been shown to promote autophagy by degrading lysosomal components, and in our previous work we showed that QC induces LC3B expression and p62 degradation, which is indicative of autophagic flux [18,20,33]. Therefore, we sought to explore a possible connection between CTSL activation and QC-induced autophagic flux. Using recombinant CTSL, we found that ectopically expressed p62 was specifically cleaved at two sites, identifying that p62 acts as a novel substrate of CTSL (Figure 5g–i, and Appendix A), mechanistically linking QC-induced CTSL upregulation to promote autophagic flux. To confirm that CTSL maintains QC-induced autophagic flux by degrading p62, we treated CTSL knockdown cells with QC and analyzed LC3B and p62 levels (Figure 5a–d). We found that p62 levels were markedly increased and LC3B levels were decreased in CTSL knockdown cells treated with QC when compared to control cells, indicating that CTSL knockdown impeded QC-induced autophagic flux (Figure 5a–h) and also that CTSL upregulation or treatment with QC increased the development of autolysosomes from early autophagosomes, which was evident from the GFP-RFP-LC3B/p62 assay (Figure 4a–i). In contrast, it was demonstrated that CTSL expression and nuclear translocation induced by the neurotoxin 6-OHDA is a mechanism that inhibits autophagy in SH-SY5Y neuroblastoma cells [41]. Also, in contrast to our finding, a recent report on periodontal ligament fibroblasts showed that CTSL upregulates p62 levels [42]. This suggested that CTSL is capable of playing opposing roles in autophagy, depending on cellular context, and implied that more work is needed to better understand the role of CTSL in autophagy. In addition, the induction of autophagy has previously been shown to activate various cathepsins, including CTSL [43], leading us to examine whether QC-induced autophagy was important for CTSL activation. In the present study we found that pharmacological shutdown of autophagy inhibited QC-induced CTSL upregulation (Figure 3c,d), indicating that QC-induced autophagy is required for QC-induced CTSL upregulation. Taken together, these data suggest that QC promotes a positive feedback loop where QC-induced autophagy upregulates CTSL, which further drives autophagic flux.

Importantly, OC cells have elevated CTSL levels compared to non-neoplastic tissues, and this elevated CTSL expression has been shown to promote ovarian cancer cell proliferation and invasion and resistance to chemotherapy [44,45], with secreted CTSL playing a role in cancer progression [46]. Additionally, inhibiting CTSL sensitizes cells to drugs such as doxorubicin, tamoxifen and so on [47]. In contrast, we demonstrated that high CTSL levels sensitized ovarian cancer cells to QC treatment. We showed that CTSL upregulation in OV2008 cells led to a marked increase in QC-induced autophagy and apoptosis, whereas CTSL KD in C13 cells decreased QC-induced autophagy and apoptosis (Figure 4, Figure 5 and Figure 6). Therefore, differences in CTSL levels could offer a therapeutic window allowing QC to preferentially target ovarian cancer cells—especially those that have developed chemoresistance—while maintaining minimal overall toxicity in vivo. This is supported by our previous work demonstrating that QC treatment sensitizes chemoresistant ovarian cancer cell lines to carboplatin treatment [18]. QC intercalates into DNA [48], suggesting that QC could directly influence CTSL expression. However, we showed that CTSL upregulation appears to be mediated by QC-induced autophagy, and more work is needed to elucidate the mechanism behind QC-induced CTSL upregulation. Although the pro- and anti-oncogenic functions of CTSL appear to be context-dependent, our work suggested that CTSL plays a role in promoting ovarian cancer cell death upon QC treatment. This anti-oncogenic function of CTSL is supported by an additional study in a mouse model of skin carcinogenesis, and CTSL is a keratinocyte-specific tumor suppressor in a mouse model of skin carcinogenesis as CTSL-deficient mice had increased oncogenic signaling accompanied by aggressive tumor behavior with increased metastasis [49]. Furthermore, CTSL released by carcinoma cells has been shown to produce the angiogenesis inhibitor endostatin from collagen XVIII, raising the possibility that CTSL could act as a putative tumor suppressor by inhibiting angiogenesis [50].

Overall, QC has previously been shown to induce cytotoxicity by a variety of mechanisms [10,47]. Here, we demonstrated that QC cytotoxicity in ovarian cancer was mediated by CTSL, and QC induced autophagic and apoptotic cell death by promoting LMP and MOMP (Figure 7b). Importantly, our work sheds light on the potential for repurposing the antimalarial QC in ovarian cancer treatment.

## 4. Materials and Methods

### 4.1. Materials

For materials refer Appendix A.

### 4.2. Cell Culture and Treatment

Human ovarian cancer cell lines HeyA8-MDR from MD Anderson (TX, USA), OVCAR5, OVCAR7, C13, OV2008 and Hek293T cells from ATCC (VA, USA) were grown in RPMI media (Gibco, Waltham, MA, USA) supplemented with 10% fetal bovine serum (R&D Systems, Minneapolis, MN, USA) and 1% penicillin–streptomycin 5% CO_2_–95% air humidified atmosphere at 37 °C and treated with QC at different concentrations (0, 2.5, 5 or 10 µM) in various experiments plus or minus specific inhibitors, including 3-methyladenine, CA-074Me, Pepstatin A, Z-FY(tBU)-DMK, Z-VAD-FMK and Bafilomycin. Cell lysates were prepared by CS lysis buffer for Western blots. All through this study, carboplatin/cisplatin-resistant C13 and HeyA8-MDR and sensitive OV2008 were used, and for the revalidation in high-grade serous OC cells, OVCAR5 and OVCAR7 cells were used. QC-induced cytotoxicity against OC cells was determined by MTT assay.

### 4.3. Generations of CTSL Knockdown and Overexpressed Stable Clones

Cell lines C13 and OVCAR5 were cultured and transfected with CTSL-sh1 (TGCCTCAGCTACTCTAACAT) sh2 (TGCCTCAGCTACTCTAACATT) and with nontargeted control shRNA as control with Lipofectamine 2000 (Invitrogen, Carlsbad, CA, USA) as per manufacturer protocol. Stable clones were selected by puromycin as reported earlier [20]. OV2008 and OVCAR7 cells were cultured and transfected with pcDNA3.1 control vector, and vector containing CTSL insert with Lipofectamine 2000 and stable clones were selected and cultured using G418. C13-ATG5 KD cells was previously reported [18].

### 4.4. Clonogenic Assay

Clonogenic survival assays were performed as previously described [51,52]. In brief, C13 and HeyA8-MDR cells were seeded at 500 cells per well in 6-well dishes in triplicate, exposed to QC (5 µM) with plus or minus specific cathepsin inhibitors including CA-074Me (CTSB inhibitor), Pepstatin A (CTSD inhibitor) and Z-FY(tBU)-DMK (CTSL inhibitor) for 24 h, rinsed with fresh medium and cultured further for 12 days. Colonies were fixed with methanol, stained with crystal violet (0.4 g/L), counted by ImageJ-Fiji version (colony count) and photographed.

### 4.5. Lysosomal Membrane Permeability/CTSL Activity Assessment

C13 and HeyA8-MDR cells were treated with 5 and 10 µM QC for 0, 3, 12, and 24 h, followed by incubation with 50 nM LysoTracker^®^ for 1 h at 37 °C and visualized by fluorescence microscopy. Cells grown in 24-well plates were treated with QC (0, 2.5, 5, and 10 µM) alone and with or without Z-FY(tBU)-DMK (10 µM) for 24 h followed by a Magic Red^®^ CTSL Assay according to the manufacturer’s protocol. LMP and CTSL activity levels were analyzed by ImageJ-Fiji version.

### 4.6. MOMP Assessment

To validate the QC effect on mitochondrial membrane potential, C13/C13 NTC, CTSL-sh1 and sh2 and OV2008 (EV and CTSL^+/+^) cells were cultured and exposed with or without QC at 5 or 10 µM and Z-FY(tBU)-DMK (10 µM) for 24 h, then incubated with 50 nM of MitoTracker^®^ Red FM for 30 min at 37 °C and visualized by fluorescence microscopy, and corrected total cell fluorescence (CTCF) was obtained by Image J.

### 4.7. Annexin V-Pacific Blue/PI Dual Staining

The Annexin V-Pacific blue/PI dual staining assay (Life Technologies) was employed to assess the QC-mediated cell death according to the protocol provided by the manufacturer. In brief, approximately 1 × 10^6^ parental (C13, HeyA8-MDR), CTSL KD (C13-NTC, sh1, sh2), and OV2008 (Vector, CTSL ^+/+^) cells were grown and treated in plus or minus combinations of QC and 10 µM specific inhibitors (Z-FY(tBU)-DMK or Z-VAD-FMK) for 24 h. The cells were sorted using a flow cytometer (BD FACS Canto II) and analyzed using Cell Quest.

### 4.8. Degradation of Cellular p62 by rCTSL or rCTSB Analysis

To assess whether p62 was directly degraded by CTSL, 50 µg of whole cell lysates from HeyA8-MDR cells were incubated with recombinant human CTSL (rCTSL) protein or rCTSB (0.0, 0.2, 0.5 and 1.0 µg/sample for 2 h or 0.5 µg for 1–3 h at room temperature) in assay buffer (50 mM MES, 5 mM DTT, 1 mM EDTA, 0.005% (w/v) Brij35, pH 6.0). Samples were resolved on a 12% SDS-PAGE gel followed by analysis of p62 degradation levels by immunoblotting with the p62 antibody. Additionally, to validate CTSL-mediated p62 cleavage pattern, lysates from Hek293T cells transfected with pDest-C-Myc-p62 construct (a kind gift of Prof.Terje Johansen, Department of Medical Biology, University of Tromsø, Tromsø, Norway) were incubated with rCTSL or rCTSB at 0.0, 0.1, 0.25 µg/sample in assay buffer and p62-myc cleavage pattern was assessed by Western blot using anti-myc antibody.

### 4.9. RNA Isolation, cDNA Synthesis and Real-Time PCR

RNAs were extracted from OV2008 and C13 cells treated with QC at 0, 5 and 10 µM using Qiagen RNA isolation kits following manufacturer’s instruction. About 1 μg of RNA was reverse transcribed using the QuantiTect Reverse Transcription cDNA synthesis kit (Qiagen, Germantown, MD, USA). Quantitative real-time PCR (qRT-PCR) was carried out using SYBR-Green PCR Master Mix (Applied Biosystems, Foster City, CA, USA) in CFX96 Real-Time PCR System (Bio-Rad, Hercules, CA, USA) as reported previously [18] by using primers synthesized from Integrated DNA Technologies (IDT). The sequences for the genes analyzed are CTSL FP:5′-CAATCAGGAATACAGGGAAGGG-3′, CTSL RP:5′-CTGGGCTTACGGTTTTGAAAG-3′, and GAPDH FP:5′-ACATCGCTCAGACACCATG-3′ and GAPDH RP:5′-TGTAGTTGAGGTCAATGAAGGG-3′.

### 4.10. Animal (In Vivo) Studies

We previously reported the inhibitory efficacy of QC on HeyA8-MDR xenografts in vivo, and the experiments were carried out under the guidelines of the Institutional Animal Care and Use Committee (IACUC) at the Mayo Foundation, in accordance with approved protocol (see the details [18]). In the current study, we analyzed the protein lysates of untreated and QC-treated xenografts from the above study by Western blot analysis.

### 4.11. Western Blot Analysis

Western blot analysis was performed as described previously [20,53]. About 30 µg of whole cell lysates from cell lines or QC-treated xenografts using cell lysis buffer from cell signaling (#9803S) were subjected to 10–15% of SDS-PAGE and transferred to nitrocellulose membrane. Western blot analysis was performed using antibodies listed in the materials and incubated with secondary anti-mouse, rabbit and/or goat-680 or 800 IR dyes and finally scanned under the Odyssey Fc Imaging system (Bio-Rad, Hercules, CA, USA). The normalized relative expression folds were calculated using ImageJ software, and values were expressed as untreated versus treated to assess fold of inhibition or activation.

### 4.12. Confocal Imaging

The HeyA8-MDR and OV2008 (EV, CTSL^+/+^) cells were grown on multi-chambered slides overnight. HeyA8-MDR cells were treated with QC at 0 and 5 µM for 24 h and OV2008 cells were exposed to Magic Red^®^ and then cells were fixed with 100% methanol. After blocking with 1% BSA in PBS, the cells were incubated at room temperature for 1 h with respective antibodies to p62 or CTSL and then washed three times with 1X PBS. p62 was detected using rabbit anti-mouse IgG Alexa Fluor^®^ 594 or rabbit anti-mouse IgG-FITC (Molecular Probes, Eugene, OR, USA and CTSL with Alexa Fluor® 488 donkey anti-goat IgG or Magic Red^®^ in 1% BSA. C13, HeyA8-MDR, OV2008 (EV, CTSL^+/+^), C13 (NTC, CTSL-sh1) cells were transfected transiently with GFP-RFP-LC3/or p62 plasmid and treated with or without QC for validating autophagic flux. Slides were mounted with Antifade reagent (Invitrogen, Carlsbad, CA, USA), visualized using a Zeiss-LSM 510 microscope and corrected total cell fluorescence (CTCF) was obtained by ImageJ-Fiji software.

### 4.13. Statistical Analysis

Data were obtained from a minimum three independent experiments in triplicates (*n* = 3). All statistical analyses were performed using GraphPad Prism version 7 software (San Diego, CA, USA). Data were analyzed using *t* test. Values were expressed using mean ± standard deviation (SD). Significance was expressed as * *p* < 0.05, ** *p* < 0.01, *** *p* < 0.001.

## 5. Conclusions

Quinacrine (QC), an antimalarial compound, has attracted great interest as a repurposed anti-cancer drug due to its ability to selectively kill tumor cells. In this study, we discovered that QC significantly upregulates cathepsin L (CTSL) but not CTSB and CTSD, implicating the specific role of CTSL in promoting QC-induced autophagic flux in ovarian cancer. QC-induced CTSL activation promotes LMP resulting in active CTSL release into the cytosol, which promotes Bid cleavage, MOMP and cyt-c release. Knockdown of CTSL markedly reduces QC-induced autophagy, LMP-MOMP; whereas ectopic expression of CTSL has an opposite effect. Additionally, using recombinant CTSL, we identified p62/SQSTM1 as a novel substrate of CTSL. Importantly we showed that upregulation of CTSL in QC-treated HeyA8-MDR xenografts corresponds with attenuation of p62, upregulation of LC3BII, and the generation of cyt-c, tBid, cleaved caspase-3 and PARP.

## Figures and Tables

**Figure 1 cancers-13-02004-f001:**
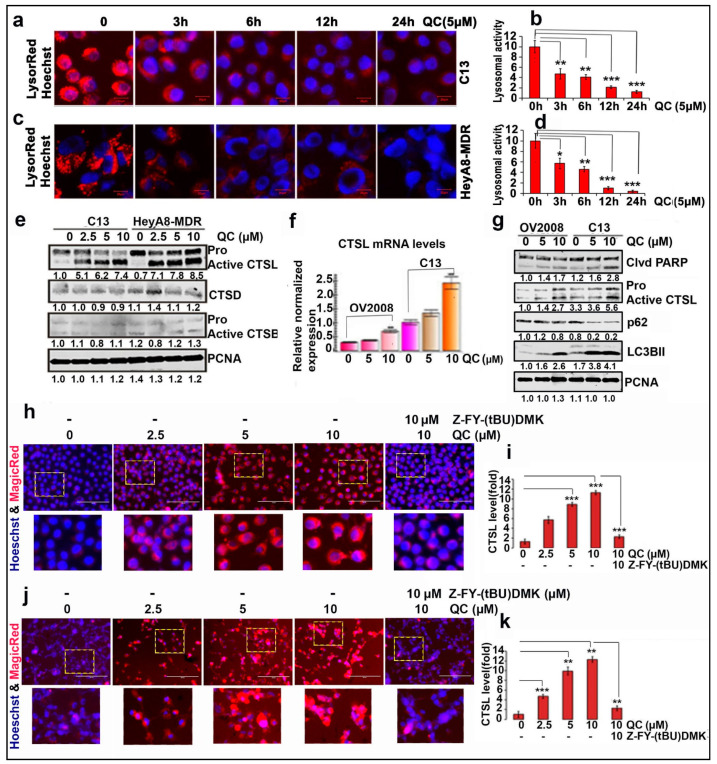
QC induces LMP by upregulating CTSL and its activity. C13 and HeyA8-MDR cells were exposed to QC at 5 µM concentration for 3, 6, 12, 24 h and labeled with LysoTracker red and Hoechst 33342, and LMP induction was assessed. Representative images of time-dependent effect of QC on LMP induction in (**a**) C13 cells and (**b**) HeyA8-MDR cells are presented. Graphical representation of QC-induced lysosomal disruption by quantifying the loss of red signal is provided for both (**c**) C13 and (**d**) HeyA8-MDR cells; (**e**) C13 and HeyA8-MDR cells were exposed to QC at 0, 2.5, 5, 10 µM concentrations for 24 h and Western analysis was performed for CTSL, CTSB and CTSD expression. (**f**) Expression of CTSL was analyzed by q-PCR upon treatment of OV2008 and C13 cells with 5 and 10 µM QC for 24 h. (**g**) Western analysis of CTSL was performed under similar treatment condition and sensitization of the cells to QC treatment was estimated by Western analysis of cleaved PARP, LC3BII and p62/SQSTM1. Magic Red assay to measure CTSL activity upon QC treatment was performed in a dose-dependent manner with or without CTSL inhibitor Z-FY (tBU)-DMK in (**h**) C13 and (**j**) HeyA8-MDR cells (Scale bar 100 µm and 40× magnification). (**i**) and (**k**) Fold change was calculated and plotted. Significance was expressed as * *p* < 0.05, ** *p* < 0.01, *** *p* < 0.001. Abbreviations: CTSL: Cathepsin-L; CTSB: Cathepsin-B; CTSD: Cathepsin-D; LC3B: Light Chain 3B; PCNA: Proliferating cell nuclear antigen.

**Figure 2 cancers-13-02004-f002:**
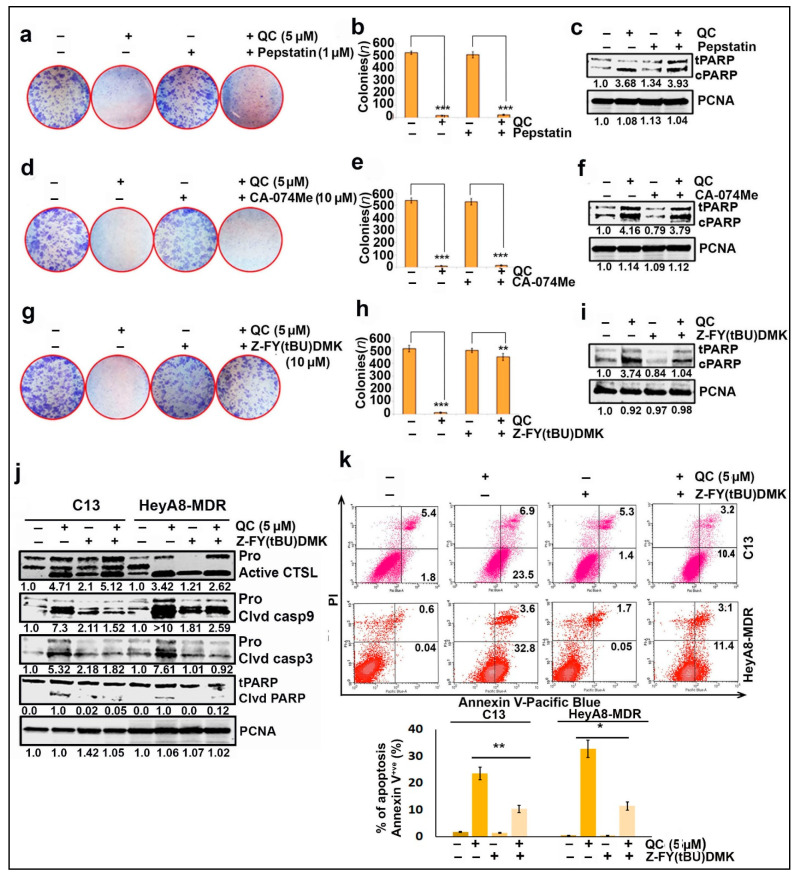
CTSL mediates QC cytotoxicity and apoptosis. Colony-forming ability was assessed in HeyA8-MDR cells where the cells were pre-treated with cathepsin selective inhibitors (**a**,**b**) Pepstatin A (1 μM, CTSD inhibitor), (**d**,**e**) CA-074 Me (10 μM, CTSB inhibitor) and (**g**,**h**) Z-FY (tBU)-DMK (10 μM, CTSL inhibitor) and exposed with or without QC (5 µM) for 24 h followed by 10 days culture. The number of colonies were counted and plotted as mean ± standard deviation. (**c,**,**f**,**i**) Western analysis was performed for cleaved PARP under the similar treatment conditions, respectively. (**j**) Levels of cleaved caspase-9/3 and cleaved PARP were assessed by immunoblot analysis upon treatment with QC in presence and absence of Z-FY (tBU)-DMK. (**k**) Annexin V/PI staining of C13 and HeyA8MDR cells treated with QC in absence and presence of Z-FY (tBU)-DMK for 24 h and FACS analysis showed QC-exhibited CTSL-mediated cell death, and the corresponding bar graph shows the percentage of Annexin-V-positive cells. Significance was expressed as * *p* < 0.05, ** *p* < 0.01, *** *p* < 0.001. Abbreviations: QC: Quinacrine; CTSL: Cethepsin L; PARP: Poly (ADP-ribose) polymerase; PCNA: Proliferating cell nuclear antigen.

**Figure 3 cancers-13-02004-f003:**
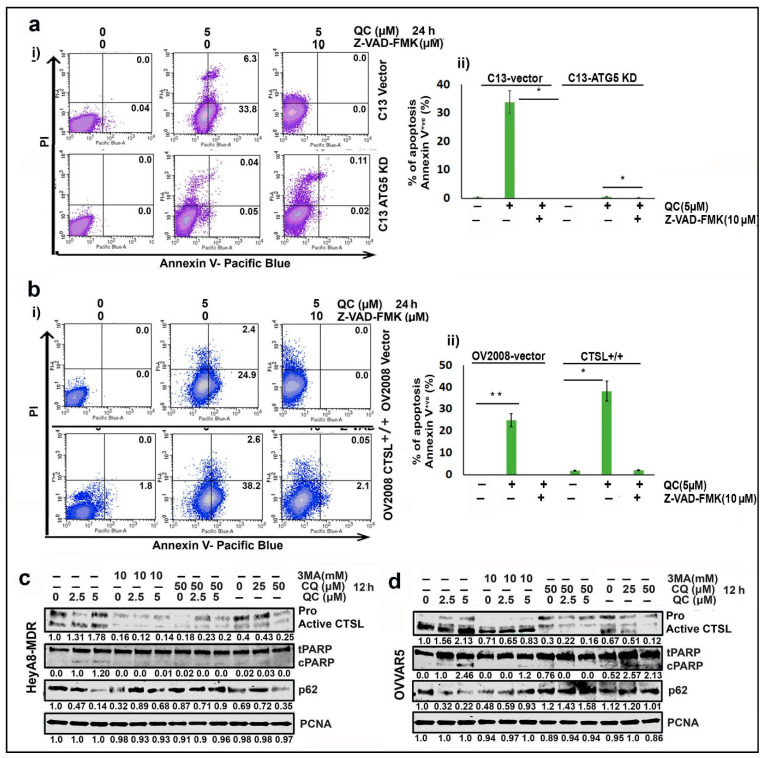
QC promotes CTSL upregulation in an autophagy-dependent manner. (**a**(**i**)) C13 ATG5 WT and KD cells were treated with 5 µM QC in the absence and presence of caspase inhibitor Z-VAD-FMK (10 μM) for 24 h and stained with Annexin V-Pacific Blue and Propidium Iodide (PI) followed by FACS analysis to determine the early and late apoptotic events, and (**ii**) the bar graph depicts percentage of Annexin-V-positive/apoptotic cells. (**b**(**i**)) Similar analysis was performed in the OV2008 cells (vector and CTSL^+/+^) and (**ii**) the pictograph shows percentage of Annexin-V-positive cells. (**c**) HeyA8-MDR and (**d**) OVCAR5 cells were treated with QC (0, 2.5, 5 μM) in the absence and presence of autophagic inhibitor 3MA (10 μM) and/or CQ (0, 25, 50 μM) for 24 h and immunoblot analysis was performed against CTSL, p62 and cleaved PARP proteins. Significance was expressed as * *p* < 0.05, ** *p* < 0.01. Abbreviations: QC: Quinacrine; 3MA: 3 Methyladenine; CQ: Chloroquin; CTSL: Cathepsin L; PARP: Poly (ADP-ribose) Polymerase; p62: Ubiquitin binding Protein 62 and PCNA: Proliferating Cell Nuclear Antigen.

**Figure 4 cancers-13-02004-f004:**
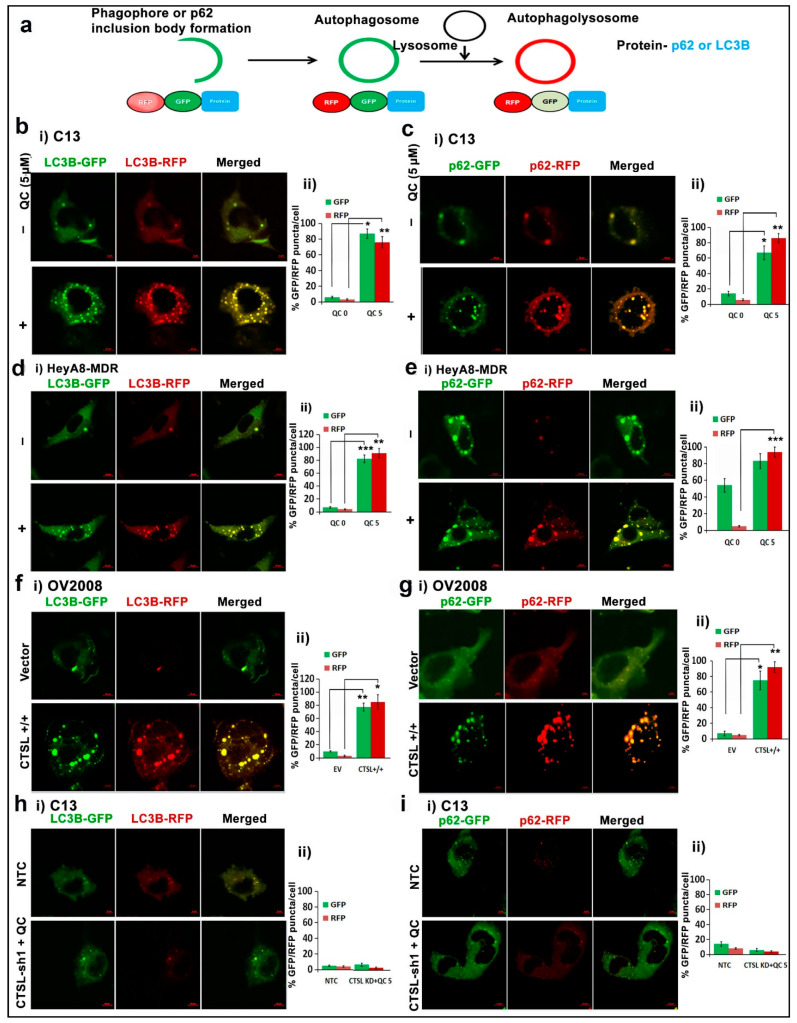
QC augments CTSL-mediated autophagic flux. (**a**) Double RFP-GFP tags were fused to LC3B or p62. QC increases autolysosomes (GFP+/RFP+ or RFP+) in (**b**(**i**) and **c**(**i**)) C13 cells transiently transfected with GFP-RFP-LC3B/or p62 plasmid, and (**b**(**ii**) and c(**ii**)) their respective pictographs shows quantification of GFP/RFP positive autolysosome puncta per cell in C13. QC augments autolysosomes (GFP+/RFP+ or RFP+) in (**d**(**i**) and **e**(**i**)) HeyA8-MDR cells transiently transfected with GFP-RFP-LC3B/or p62 plasmid, and (**d**(**ii**) and **e**(**ii**)) their respective pictographs shows quantification of GFP/RFP positive autolysosome puncta per cell in HeyA8-MDR. Similar analysis was performed in the (**f**(**i**) and **g**(**i**)) OV2008 cells (vector and CTSL^+/+^) and (**h**(**i**) and **i**(**i**)) C13 (NTC, sh1) and (**f**(**i**)–**i**(**i**)) their respective bar graphs depict quantification of GFP/RFP positive autolysosome puncta per cell in OV2008 and C13 cells, respectively (scale bar 10 µm and 40× magnification). Significance was expressed as * *p* < 0.05, ** *p* < 0.01, *** *p* < 0.001. Abbreviations: CTSL: Cathepsin-L; LC3B: Light Chain 3B; p62: Ubiquitin binding Protein 62 and RFP/GFP: Red/green fluorescent protein.

**Figure 5 cancers-13-02004-f005:**
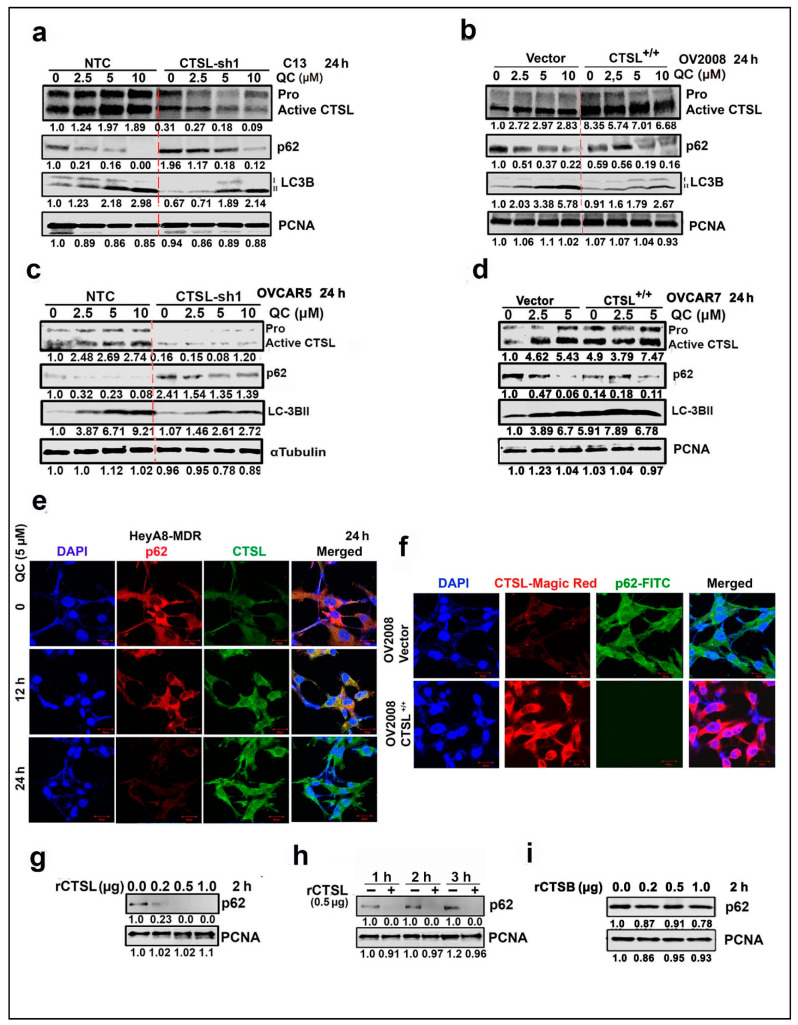
QC-upregulated CTSL drives the autophagic flux by degrading p62. (**a**,**c**) Western analysis in the CTSL KD sh1 clone of C13 and OVCAR5 cells upon treatment with 2.5, 5 and 10 μM QC for 24 h against CTSL, LC3BII and p62 proteins. (**b**,**d**) Similar immunoblot analysis in the CTSL-overexpressed OV2008 and OVCAR7 cells for the mentioned proteins. (**e**) Confocal analysis for CTSL (green) and p62 (red) was performed in the HeyA8-MDR cells upon treatment with QC for 12 and 24 h (Scale bar: 20 μm and 40× magnification). (**f**) Magic Red staining for CTSL activity assay and p62 labeling was done in the OV2008 control transfected and the CTSL-overexpressed cells, and representative images of confocal analysis are provided (Scale bar: 20 μm and 40× magnification). (**g**) Western analysis against p62 protein was performed in the HeyA8-MDR cells upon incubation with rCTSL at 0.0, 0.2, 0.5 and 1.0 µg/sample for 2 h. (**h**) Similar Western analysis was performed upon treatment with 0.5 µg rCTSL for 1, 2 and 3 h, respectively. (**i**) Western analysis against p62 protein was performed in the HeyA8-MDR cells upon incubation with rCTSB at 0.0, 0.2, 0.5 and 1.0 µg/sample for 2 h. Abbreviations: QC: Quinacrine; CTSL: Cathepsin-L; LC3B: Light Chain 3B; p62: Ubiquitin binding Protein 62 and and PCNA: Proliferating Cell Nuclear Antigen.

**Figure 6 cancers-13-02004-f006:**
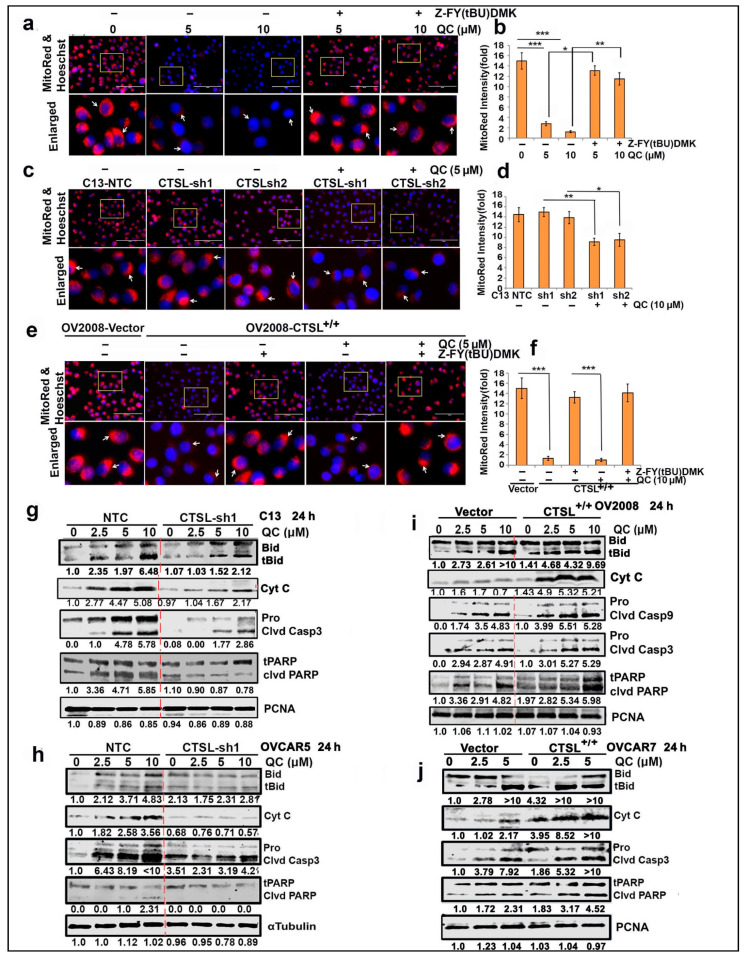
CTSL mediates QC-induced MOMP and cleavage of Bid, caspase-3 and PARP in ovarian cancer cells. (**a**) C13 treated with QC (0, 5, 10 µM) in the presence and absence of Z-FY(tBU)DMK (10 µM) were labeled with Mitotracker Red and Hoechst 33342 for nuclei, and MOMP induction was assessed using confocal microscopy. Representative images are presented. (**b**) Quantitation of the signal is presented in a bar diagram. A similar study was performed with (**c**) C13 NTC and sh1/sh2 cells and (**e**) OV2008 vector transfected and CTSL-overexpressed cells upon treatment with 5 µM QC in plus or minus Z-FY(tBU)-DMK for 24 h, and representative images are given. (**d**,**f**) Graphical representation of the MOMP induction. (**g**,**h**) Western analysis was performed in CTSL KD and NTC control transfected C13 and OVCAR5 cells upon treatment with indicated concentrations of QC for 24 h against the proteins tBid, cyt-c, cleaved caspase 3 and cleaved PARP. (**i**,**j**) Similar Western analysis was carried out for control vector and CTSL-overexpressed OV2008 and OVCAR7 cells. Scale bar 200 µm and 40× magnification and Significance was expressed as * *p* < 0.05, ** *p* < 0.01, *** *p* < 0.001. Abbreviations: QC: Quinacrine; CTSL: Cathepsin-L; tBid: truncated Bcl-2 family member(Bid); Cyt c: Cytochrome-c; Casp: Caspase; PARP: Poly (ADP-ribose) polymerase.

**Figure 7 cancers-13-02004-f007:**
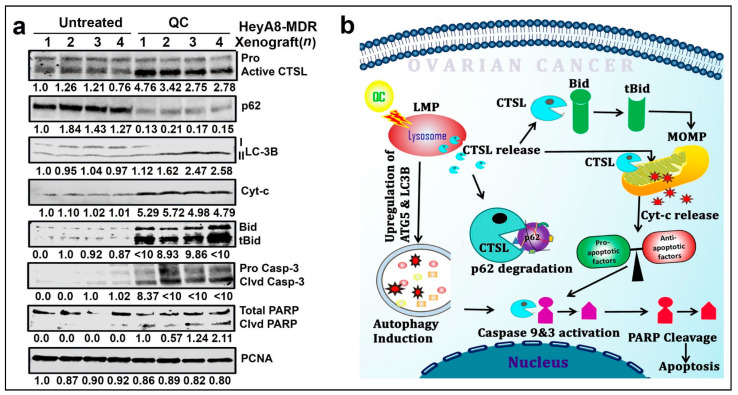
QC induces CTSL and suppresses tumor growth in the HeyA8-MDR xenograft model. (**a**) Immunoblot analysis was performed in the untreated and QC-treated HeyA8-MDR OC xenograft tumors against CTSL, p62, LC3BII, tBid, Cyt-c, cleaved PARP and cleaved caspase 3 expressions. (**b**) Schematic representation of the study. Abbreviations: QC: quinacrine; CTSL: Cathepsin-L; LMP: Lysosomal Membrane Permeabilization; MOMP: Mitochondrial Outer Membrane Permeabilization; tBid: truncated Bcl-2 family member(Bid); p62- ubiquitinbinding protein 62; Cyt c: Cytochrome-c; ATG5: Autophagy Related 5; LC3B: Autophagy marker protein Light chain 3B; PARP: Poly (ADP-ribose) polymerase.

## Data Availability

Data are contained within the article or Appendix A.

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
