# Peer review of "Quinacrine-Induced Autophagy in Ovarian Cancer Triggers Cathepsin-L Mediated Lysosomal/Mitochondrial Membrane Permeabilization and Cell Death"

_cancers, 2021, doi:10.3390/cancers13092004_

Round 1

Reviewer 1 Report

In this paper the authors investigate mechanisms underlying ovarian cancer cell death upon treatment with Quinacrine. They find that Quinacrine induces autophagy that triggers Cathepsin-L-mediated lysosomial/mithocondrial membrane permeabilization leading to apoptotic cell death in ovarian cancer cell lines and in xenograft model, stressing the importance of Quinacrine as anticancer agent.

The article is very well written and the experiments are convincing so I just have some minor revisions:

-There are several western blots in the figures but the authors only show densitometric analysis of one experiment so it seems they have done every experiments just one time. I think they should plot densitometric analysis of every western blot on a graph (Histograms should represent the mean plus SD of the densitometric analysis of the ratio of specific protein/PCNA of three experiments).

-Some figures are “overcrowded”, in Fig. 5 and 6 they show silencing with two different shRNA, I think one is enough (the other can be moved to supplementary).

Author Response

Reviewer 1: In this paper the authors investigate mechanisms underlying ovarian cancer cell death upon treatment with Quinacrine. They find that Quinacrine induces autophagy that triggers Cathepsin-L-mediated lysosomial/mithocondrial membrane permeabilization leading to apoptotic cell death in ovarian cancer cell lines and in xenograft model, stressing the importance of Quinacrine as anticancer agent.

The article is very well written and the experiments are convincing so I just have some minor revisions:

-There are several western blots in the figures but the authors only show densitometric analysis of one experiment so it seems they have done every experiments just one time. I think they should plot densitometric analysis of every western blot on a graph (Histograms should represent the mean plus SD of the densitometric analysis of the ratio of specific protein/PCNA of three experiments).

Response: Thanks for the valid comment. All the experiments were performed in triplicates as mentioned in statistics section. Since we used several and long blots for this study, inserting graphs will make figures overcrowded. Hence, average mean values were presented in all western blot results to avoid crowed figures. However, we made densitometric graphs for all the blots with SD and submitted as supplementary figures and incorporated appropriately in results sections.

-Some figures are “overcrowded”, in Fig. 5 and 6 they show silencing with two different shRNA, I think one is enough (the other can be moved to supplementary).

Response: CTSL-sh2 panels in fig.5 and 6 were shifted to supplementary sections (supplementary figure 3).

Reviewer 2 Report

The study "Quinacrine-induced autophagy in ovarian cancer triggers Cathepsin-L mediated lysosomal/mitochondrial membrane permeabilization and cell death" is very interesting because examines mechamisms of quinacrine-induced autophagy in ovarian cancer.

This manuscript is well written and results are very well and clear presented

Author Response

Thanks for your appreciation on our research work.

Reviewer 3 Report

The manuscript by Thirusangu et al, using drug repurposing approach provides detailed mechanism of the antimalarial compound quinacrine (QC) in induction of autophagy by upregulating cathepsin L (CTSL) and ultimately inducing in cell death in ovarian cancer cells. The manuscript is well written and provide enough evidence of molecular mechanism of action of quinacrine in ovarian cancer. However, few minor concerns are noted which are mention below-

  • Authors are advised to show toxicity of QC (e.g. Cytotoxicity assay) in various OC cell lines at different time point.
  • The fluorescence image provided in fig. 1h and 1j is not clear, authors are advised to provide clear image for the same.
  • Authors are advised to show percentage of apoptosis in the form of bar graph for the fig. 2, fig. 3a and 3b.
  • It will easier for the readers to understand the in vivo effect of QC on OC, if authors can provide the tumor weight and volume data of mouse-derived xenograft of various OC cells after treatment with QC.

Author Response

Reviewer 3: The manuscript by Thirusangu et al, using drug repurposing approach provides detailed mechanism of the antimalarial compound quinacrine (QC) in induction of autophagy by upregulating cathepsin L (CTSL) and ultimately inducing in cell death in ovarian cancer cells. The manuscript is well written and provide enough evidence of molecular mechanism of action of quinacrine in ovarian cancer. However, few minor concerns are noted which are mention below-

Authors are advised to show toxicity of QC (e.g. Cytotoxicity assay) in various OC cell lines at different time point.

Response: Cytotoxicity were analyzed in different ovarian cancer cell lines used for this study and presented as supplementary information (supplementary fig 1a and b).

The fluorescence image provided in fig. 1h and 1j is not clear, authors are advised to provide clear image for the same.

Response: Images were enlarged for better quality and inserted in Figure 1.

Authors are advised to show percentage of apoptosis in the form of bar graph for the fig. 2, fig. 3a and 3b.

Response: Thanks for your valuable suggestion, graphs for percentage of apoptosis for Fig 2, Fig3a and 3b inserted as per reviewer suggestion.

It will easier for the readers to understand the in vivo effect of QC on OC, if authors can provide the tumor weight and volume data of mouse-derived xenograft of various OC cells after treatment with QC.

Response: QC treatment of mouse-derived xenografts of HeyA8-MDR cells was shown to be effective in reducing tumor weight and ascitic fluid formation (refer figure 7 in Ashwani et al,2015 [18]). For current study, as a continuation work, tumor tissues from untreated and QC treated groups used to investigate molecular mechanism underlying QC induced autophagy/CTSL dependent LMP/MOMP in in vivo. Same was mentioned in in-vivo result section.